# [Proposal-ML]: Models Supporting Long Time Series

**Max Xu**
2024403086
maxzixiaoxu@gmail.com

## Abstract

This study explores basic models supporting long time series tasks. By creating baseline models while building and maintaining time-series data, we look to research improvements to time complexity of transformers with Mamba as the backbone for the model training. In this study we inspect mechanisms and techniques to handle long time series.

## 1 Background

In recent years, the fields of machine learning have witnessed advancements, particularly through the adoption of architectures such as transformers. These models have revolutionized the way we understand and handle data sequences, enabling performance on a range of tasks. However, despite their success, transformers face challenges related to computational efficiency, particularly in scenarios involving large datasets and real-time applications.

Transformers and Mamba architecture have a range of applications, notably being in natural language processing, revolutionizing understanding and generation of human language. With data sequences such as text, Mamba allows dynamic weighting of data. In the context of long sequences, transformers are unable to store large data providing importance to analyzing methods to capture long time series. This will also contribute to the general field of time series forecasting.

## 2 Related Work

Mamba, an emerging framework, utilizes Selective Structured State Space Model. It leverages parallelization and resource management, providing a practicality compared to square limit time complexity of the transformer architecture. It is ideal for time-series forecasting as it can capture long-term dependencies through data selectivity. The Selective State Space uses four selective parameters with time and dimension, effectively allowing for dynamic processing of the data.

An existing approach and analysis of Mamba suggests although it has advantages for short-context NLP tasks, it has limited comprehension for long-contexts compared to transformer models (Yuan et al. 2024). The study proposes the use of ReMamba, incorporating selective compression and selective adaption to attempt for an advanced Mamba. The further selectivity produces more effective key information retention and reduces state space update frequency. In the experiment, the parameters for the hidden state of the Mamba model are manipulated with compression ratio $\rho$, relative length to compress p, and the relative start position s. The experimental results show compressing from the start of the sequence with s = 0 resulted in the optimal model.

Preprint. Under review.

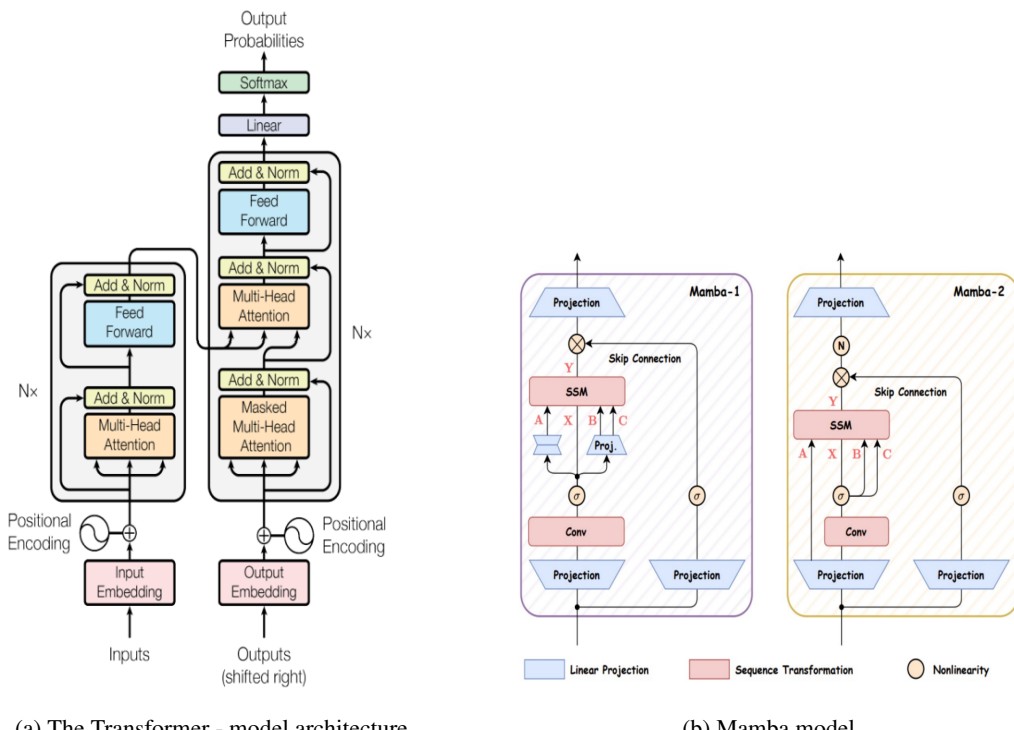

(a) The Transformer - model architecture          (b) Mamba model

Figure 1

| Model | $\rho$ | $p$ | $s$ | model_type | average |
|---|---|---|---|---|---|
| ReMamba | 0.009 | 0.18 | 0.00 | ReMamba | 27.86 |
| middle0.0 | 0.009 | 0.18 | 0.00 | middle | 25.96 |
| middle0.1 | 0.009 | 0.18 | 0.10 | middle | 26.45 |
| middle0.2 | 0.009 | 0.18 | 0.20 | middle | 26.86 |
| middle0.3 | 0.009 | 0.18 | 0.30 | middle | 26.56 |
| middle0.4 | 0.009 | 0.18 | 0.40 | middle | 26.43 |
| special | 0.009 | 1.00 | 1.00 | special | 15.76 |

Figure 2: Remamba results

## 3    Proposed Method

### 3.1    Motivation

One necessity for long-context tasks is the expansion of the context window. Transformers are limited by space when considering contextual information. The motivation is to examine non-transformer architecture basic time series models and methods with the motivation to improve performance and handling capabilities of long time series. I will also examine the zero-shot learning process in the model. By systematically evaluating the performance of transformers within the Mamba framework, I hope to contribute valuable insights for future use in time-series forecasting applications.

### 3.2    Method

A group at Tsinghua University plan to build a Time-Series-Repo for training. I propose to work with them to observe the data and performance results to identify bottlenecks and determine potential developments to long time series tasks.

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
