# OpenReview forum: "[Proposal-ML] Models Supporting Long Time Series"
_tsinghua.edu.cn/THU/2024/Fall/AML — THU 2024 Fall AML Submission_

### Official Review · ~Tianxing_Yang1 · 2024-11-11
**Review for Proposal on Exploring Basic Model Architectures for Long Sequence Tasks**

**Rating:** 7
**Confidence:** 3

**Review:**

This proposal aims to explore basic model architectures that support long-sequence tasks.

Pros:

- The comparison between Transformer and Mamba models, as well as the design of a basic architecture, is a fundamental topic in the field of large language models (LLMs).
- The proposal is clearly written and focuses on addressing the issue of Mamba architecture’s performance limitations when handling long text sequences.

Cons:

- The proposal outlines the problem it aims to solve but does not discuss the specific methods or strategies planned to address this issue.

Minor Issue:

- The citation format in the proposal is not correctly applied.

---

### Official Review · ~Zihan_Lv1 · 2024-11-11
**Good exploration with Practical Significance Potential**

**Rating:** 7
**Confidence:** 3

**Review:**

This project focuses on solving the computational complexity problem of Transformer in processing long sequences in response to the practical needs of long time sequence processing, and plans to collaborate with other research teams to facilitate the acquisition of the training data required for the study. However, more empirical studies are needed since it is still controversial whether Mamba has better performance than existing studies.

---

### Official Review · ~Fabian_Pawelczyk1 · 2024-11-11
**Good proposal**

**Rating:** 7
**Confidence:** 4

**Review:**

**Decision: Accept**

This proposal presents a practical approach to improving the efficiency of time-series models, focusing on the Mamba architecture to address transformers' limitations with long data sequences. The problem is well-defined, recognizing that transformers struggle with extended time sequences due to high computational requirements. The solution—a selective data-processing approach using Mamba—seems promising for time-series forecasting, and the proposal outlines a thoughtful direction for enhancing data retention and processing efficiency.

Strengths of the proposal include the author’s use of both established Mamba techniques and the development of ReMamba, which seeks to compress data selectively for better performance. The planned collaboration with a Tsinghua University team on a Time-Series Repository could add valuable insights and provide a solid framework for testing. This practical setup adds depth to the project, though the proposal would benefit from clearer plans for evaluating success metrics.

---

### Official Review · ~Anqi_LI5 · 2024-11-11

**Rating:** 8
**Confidence:** 3

**Review:**

The proposal presents a clear and focused research direction, addressing the challenges of long time series tasks and exploring potential solutions using Mamba architecture. The proposal is well-structured and organized, with clear headings and subheadings. The abstract succinctly summarizes the research objectives and methodology.
Pros:
Clear research direction: The proposal focuses on addressing the limitations of transformers for long time series tasks and exploring alternative approaches.
Practical methodology: The proposed collaboration with the Tsinghua University group to build and analyze a Time-Series-Repo shows a practical approach to gathering data and evaluating model performance.
Potential for impact: The research has the potential to significantly improve the performance and handling capabilities of long time series tasks, with applications in various fields.
Cons:
Limited details on specific models: The proposal mentions exploring non-transformer models and Mamba architecture but lacks specific details on the models and methods to be used.
Lack of evaluation metrics: The proposal does not mention specific evaluation metrics to measure the performance of the models.
Potential challenges with collaboration: Collaborating with another research group could present challenges in terms of coordination and data sharing.

---

### Official Review · ~Ziyu_Zhao6 · 2024-11-11
**Review of "Models Supporting Long Time Series" Proposal**

**Rating:** 7
**Confidence:** 3

**Review:**

This project aims to enhance time-series forecasting methods by integrating the Mamba architecture with transformer models, specifically focusing on long-context tasks where transformers face computational limitations. Overall, the proposal is easy-to-read, and well-organized. Although this study explored the application of the Mamba architecture to long-sequence tasks, its innovation is limited. In addition, these improvements may not be applicable to a wide range of fields with different data characteristics, limiting their versatility. The high dependence on parameters increases complexity, and the dependence on experimental results makes its performance in practical applications uncertain. Overall, this study has made some breakthroughs in the idea of ​​processing long sequences but lacks innovation and is mainly an enhancement of existing technologies.

---

### Official Review · ~Xin_Chen65 · 2024-11-11
**Good proposal**

**Rating:** 8
**Confidence:** 3

**Review:**

This proposal presents a comprehensive approach to addressing the challenges of handling long time series data within the realm of machine learning.

strength: (1) The proposal tackles a significant issue in the field of machine learning, which is the computational efficiency of transformers when dealing with long time series data. The introduction of the Mamba architecture as a backbone for model training is a novel approach. (2)  The proposal demonstrates a deep understanding of the transformer model's limitations and the Mamba architecture's capabilities. The discussion on the Selective Structured State Space Model and its advantages over traditional transformers is well-articulated.

weakness: (1) The proposal mentions the examination of the zero-shot learning process within the model but does not elaborate on how this will be implemented or its expected impact. Providing more details on this aspect would strengthen the proposal. (2) The proposal mentions the manipulation of parameters for the Mamba model but does not provide details on the experimental design, such as the dataset used, the evaluation metrics, or the baseline models for comparison.

---

### Official Review · ~Zheng_Jiang2 · 2024-11-11
**An interesting exploration for long time series tasks**

**Rating:** 7
**Confidence:** 4

**Review:**

This proposal outlines a study on basic models for supporting long time series tasks, with a focus on improving the time complexity of transformers using the Mamba architecture as the backbone for model training. The paper provides a good overview of related work, including the capabilities and limitations of Mamba and transformer models in handling long sequences.

However, I have some questions or suggestions for it:
1. The paper lacks detailed information on the specific methodologies and techniques that will be used to improve the handling of long time series data.
2. The proposal seems to rely heavily on the Mamba architecture, with less emphasis on other potential approaches or architectures that could complement or enhance the model.

---

### Official Review · ~Zhuofan_Sun1 · 2024-11-12

**Rating:** 10
**Confidence:** 5

**Review:**

Strengths:

Focus on Long Time Series: The proposal addresses a critical gap in time series analysis by focusing on models that can effectively handle long sequences. This is essential for many real-world applications where long-term dependencies are crucial.
Exploration of Non-Transformer Models: While transformers have been successful, exploring alternative architectures is valuable. The proposal mentions considering non-transformer models, which could lead to new insights and potentially better performance for long time series tasks.
Collaboration with Tsinghua University: The collaboration with Tsinghua University to build a Time-Series-Repo is commendable. This data resource will be invaluable for training and evaluating models, and it will facilitate reproducibility and further research in the field.
Evaluation of Mamba and ReMamba: The proposal plans to evaluate the performance of Mamba and ReMamba within the Mamba framework. This will provide valuable insights into their effectiveness for long time series tasks and help identify potential improvements.
Overall Impression:

The proposal presents a promising approach for handling long time series tasks using Mamba architecture. The focus on long sequences and the exploration of non-transformer models are valuable contributions to the field. However, the proposal could benefit from more detail on the proposed method, inclusion of baseline models for comparison, and further explanation of the role of zero-shot learning. With these additions, the proposal has the potential to lead to significant advancements in long time series analysis.

---

### Official Review · ~Killian_Conyngham1 · 2024-11-12
**Review for Models Supporting Long Time Series**

**Rating:** 8
**Confidence:** 4

**Review:**

This is a clear and important proposal. A strong case is made for the potential of applying Mamba as the backbone for training long time series models. The related work section gives a clear overview of Mamba, and the newer developments of ReMamba. While the case for the potential of this approach is well-made, additional details regarding the exact nature of the project would be of great benefit. Specifically, the nature of collaboration and use of the Time-Series-Repo and the scope of this project in particular could be more clearly defined.

---

### Official Review · ~Jackson_M_Luckey1 · 2024-11-12
**Proposal Review**

**Rating:** 8
**Confidence:** 3

**Review:**

The proposal is quite interesting, but it is a little unclear how the MAMBA time series model will be evaluated and what it will be compared to. The related work section provided a solid overview. I appreciate including the figures that explain the basic concepts behind MAMBA.

---

### Official Review · ~Zhu_Zhang6 · 2024-11-12
**Good proposal, more work should be done**

**Rating:** 7
**Confidence:** 3

**Review:**

**Summary:**

This proposal explores the use of Mamba architecture to support long time series tasks, aiming to address limitations in transformer models regarding computational efficiency and contextual handling of long sequences. The study highlights Mamba’s linear time complexity and selective state space model, which allow it to handle long-term dependencies effectively. The proposal intends to collaborate with a team at THU to build a time-series repository, evaluate the performance of Mamba in various long-context scenarios, and experiment with techniques like selective compression to enhance Mamba's handling of extensive sequences.

**Strengths:**
1. **Focus on Long-Sequence Efficiency:** By choosing Mamba, the study targets a known limitation of transformers, providing an efficient alternative for time-series forecasting and analysis of long data sequences.

**Weaknesses:**
1. **Limited Details on Evaluation Metrics:** There is minimal information on specific metrics that will assess Mamba’s performance, limiting the clarity on how success will be measured.

**Questions:**
1. Will there be comparisons with other non-transformer models to establish the effectiveness of Mamba in this domain?
2. Are there plans to explore additional architectures or hybrid models beyond Mamba and ReMamba to achieve even greater efficiency?